# Gut microbiome and plasma metabolome alterations in ileostomy and after closure of ileostomy

Liang Xu,[1,2] Xiaolong Li,[1] Lang Chen,[1] Haitao Ma,[1] Ying Wang,[1] Wenwen Liu,[3] Anyan Liao,[3] Liang Tan,[4] Xiao Gao,[5] Weidong Xiao,[1] Hua Yang,[6] Guangyan Ji,[3] Yuan Qiu[1]

**ABSTRACT**  A temporary loop ileostomy is a routine procedure for protecting the anastomosis in patients undergoing radical resection of rectal cancer. Fecal diversion by a diverting ileostomy may induce microbiota dysbiosis in the defunctioned colon; however, data on temporal and spatial microbiome and metabolome changes in these patients are sparse. Thirty patients who underwent ileostomy closure were enrolled. Fecal and plasma samples were collected successively before ileostomy closure, at the first postoperative defecation, and 1 month postoperatively. The 16S rRNA gene sequencing was used to assess changes in gut microbes, and metabolic components in the plasma were analyzed using global untargeted metabolomics. Advanced data analysis methods were used to examine the differences and correlations between flora and metabolites. The gut microbiota in the ileostomy effluent and defunctioned colon had lesser species diversity and richness, with an abundance of aerobic, gram-negative, and potentially pathogenic bacteria. After the intestinal continuity was restored with routine meal feeding, the gut microbes recovered to a standard composition within 1 month. Moreover, xanthine, traumatic acid, L-glutamine, and norepinephrine levels increased markedly in patients with ileostoma. The ileostomy closure reversed the ileostomy-associated metabolic alterations, including an increased abundance of L-leucine, creatine, and 2-ketobutyric acid. Furthermore, *Agathobacter* and *Peptostreptococcus* were most closely associated with the reconstruction of postoperative gut microbes. We described a spatiotemporal map of the intestinal microbial ecological reconstruction and metabolic recovery before and after ileostomy reversal for perioperative intervention in patients with ileostomy closure surgery.

**IMPORTANCE**  In this paper, the changes in the intestinal microbiome and plasma metabolome before and after temporary ileostomy were reported for the first time, and the dynamic changes in intestinal contents were described. At the same time, the key bacterial genera involved in the reestablishment of microflora after the restoration of intestinal continuity were found, and the key relationship between them and plasma metabolites was also found. More importantly, we found that patients with ileal fistula may be at risk of metabolic imbalance and that this particular metabolic state may potentially affect the course of tumor treatment. Finally, the samples in this study were obtained in their natural state and can be easily applied to the clinic to avoid unnecessary invasive examinations.

**KEYWORDS**  temporary ileostomy, gut microbes, plasma metabolites, colorectal cancer

Temporary loop ileostomies are often used to protect low colorectal anastomoses (1). Ileostomy reversal was scheduled for several months after the primary surgery (radical resection of rectal cancer plus ileostomy) when the integrity of the primary anastomosis was ensured (2). After ileal fecal diversion, the distal defunctioned intestine

**Peer Reviewers** Qin Liu, The Chinese University of Hong Kong, Hong Kong, Hong Kong; Song Liu, Nanjing Drum Tower Hospital, Nanjing, China; Yang Yang, Nanjing University Medical School, Nanjing, China

Address correspondence to Guangyan Ji, jiguangyan168@163.com, or Yuan Qiu, xiaoq2037@qq.com.

Liang Xu, Xiaolong Li, and Lang Chen contributed equally to this article. The author order was based on the time of their participation in the research project.

The authors declare no conflict of interest.

See the funding table on p. 16.

is deprived of fecal stream stimulation, including digestive fluid, nutrients, microbes, and their metabolites, for several months. The colons of healthy individuals harbor an enormous number of bacteria in mutualistic symbiosis with the host. The colonic tract provides survival conditions for the microbiota, which is considered a new metabolic organ involved in regulating host metabolism (3). Once an ileostomy is formed, the distal colon loses its ability to store the stool and microbiota. A considerable proportion of patients who live with this stoma are treated with antitumor therapy. Therefore, their metabolic statuses warrant further study.

Microbiome composition is relatively stable in healthy individuals after successive waves of colonization in early life (4, 5). However, completely diverting ileostomy disturbs colonic homeostasis (6, 7). As the patient recovers from the ileostomy reversal surgery several months later, colon homeostasis can be re-established. Accumulating evidence has indicated a link between certain gut microbial species and circulating metabolites (3). Studies on these patients may provide valuable information on the plasma metabolome that characterizes patients with and without ileostomy, especially by linking information on the stool microbiome with blood metabolite data.

In this study, we performed deep 16S rRNA amplicon gene sequencing and plasma metabolomic profiling in a cohort of patients before and after ileostomy reversal. This longitudinal multi-omic approach revealed the differences and associations between individual fecal microbial communities and circulating metabolites before and after ileostomy reversal.

## MATERIALS AND METHODS

### Study participants screening and sample collection

Thirty patients who underwent laparoscopic anterior resection of the rectum with ileostomy, followed by ileostomy closure, at the Second Affiliated Hospital of the Army Medical University from November 2020 to October 2021 were enrolled. These samples and clinical information were collected with informed consent and with the approval of the corresponding institutional review board. Ileostomy closure was scheduled for 1–20 months after the primary surgery (anterior resection of the rectum plus ileostomy) once the integrity of the primary anastomosis was ensured. Patients who received antitumor therapy within 1 month prior to the first sample collection and during follow-up or developed intestinal complications after the closure surgery, such as intestinal obstruction and fistula, were excluded.

Samples were collected at three time points during the perioperative period. Time point 0 (E0) was before ileostomy closure; E0 fecal samples were collected from the small bowel effluent before ileostomy closure. Time point 1 (E1) was the time of the first defecation after the closure surgery, while time point 2 (E2) was 1 month after the ileostomy closure. Both E1 and E2 fecal samples were collected from the anus. Venous blood samples were drawn after an overnight fast. The samples were collected in an airtight container and placed directly on dry ice and then stored at −80°C for 16S rRNA gene sequencing and metabolomic analyses. Figure 1 presents the entire process and its groupings.

### Fecal DNA extraction and 16S rRNA gene sequencing

Microbial DNA was extracted from fecal samples using the HiPure Stool DNA Kit (Magen, Guangzhou, China) following the manufacturer's protocols. The 16S ribosomal DNA target regions were amplified by polymerase chain reaction (PCR) using primers targeting the V3 and V4 regions of the 16S rRNA gene (8). The reagents used to perform the PCR reaction, including reaction buffer, dNTPs, high GC enhancer, high-fidelity DNA polymerase, each primer, and template DNA are manufactured by New England Biolabs (USA). Following the manufacturer's instructions, we extracted the amplicon from 2% agarose gels, then purified it with an AxyPrep DNA Gel Extraction Kit (Axygen

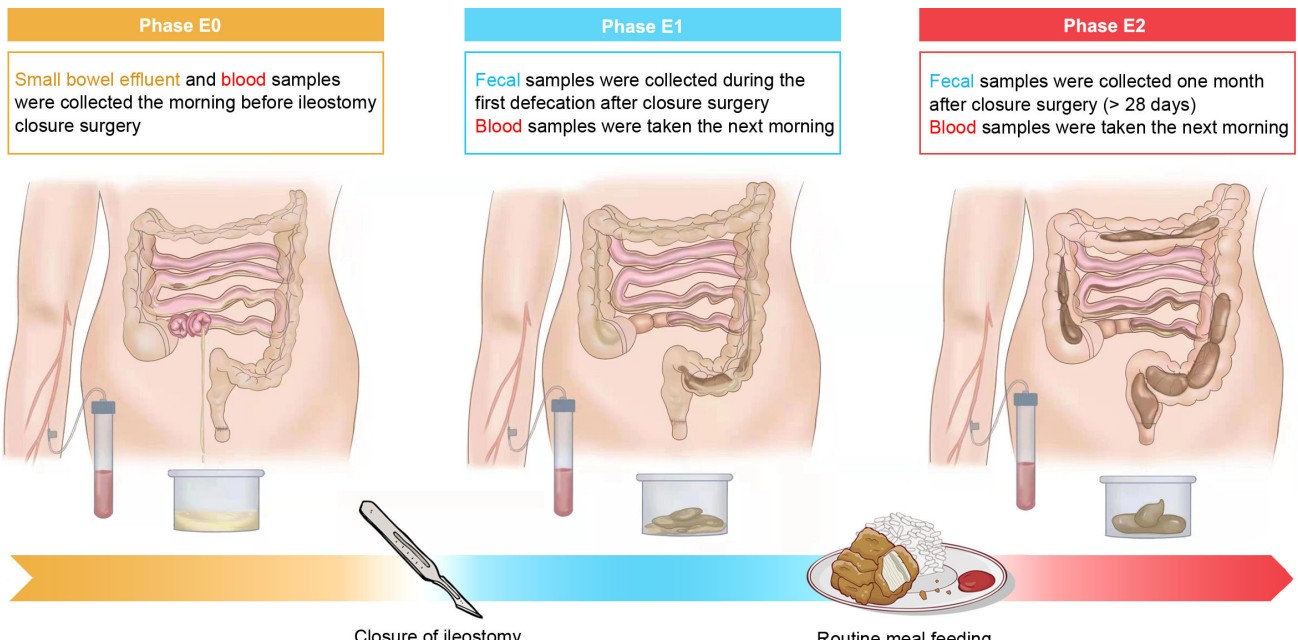

**FIG 1** Grouping of participants and flowchart. The flowchart details the disposal methods, sample types, and collection time points of phases E0, E1, and E2. Groups are marked with corresponding colors.

Biosciences, Union City, CA, USA), and finally quantified it with an ABI StepOnePlus Real-Time PCR System (Life Technologies, Foster City, CA, USA). Purified amplicons were pooled in equimolar amounts and paired-end sequenced (PE250) on an Illumina platform according to standard protocols. Raw reads were deposited in the Sequence Read Archive (SRA) database of the National Center of Biotechnology Information (NCBI).

## Quality control, clustering, and taxonomy annotation of microbial sequencing

After obtaining raw sequencing reads, low-quality reads were filtered using the FastP software (v.0.18.0) and then assembled (9). Double-ended reads were spliced into tags, which were then filtered using the FLASH software (v.1.2.11) (10). The resulting data were referred to as a clean tag (11). Clustering was then performed using the clean tag, and the chimeric tag detected during the clustering process was removed using the USEARCH software (v.9.2.64) (12). The sequences obtained were effective tags with >97% similarity and classified as operational taxonomic units (OTU). While constructing OTUs, representative sequences were selected and annotated with the BLAST database (v.2.6.0) (13) using the naive Bayesian assignment algorithm of the ribosomal database project (RDP) classifier (v.2.2, the confidence threshold was set at 0.8) (14).

### Microbial sequencing data analysis

In the gut microbiome composition analysis, we plotted stacked bar (15) and circular layout charts (16) and conducted Pearson correlation analysis of species (17). The Circlize package in R was used for data acquisition and Circos diagram rendering. We then plotted the OTU/ASV (amplicon sequence variants) rarefied curve and rank abundance curve (15) and calculated Chao1, Sob, Shannon, and Simpson indices (18) to compare the α diversity of microorganisms. We also compared the β diversity, and sequence alignment was performed first (19). Subsequently, principal coordinate analysis (PCoA) and non-metric multidimensional scaling analysis (NMDS) models were used to draw a two-dimensional multivariate statistical analysis diagram, and the ANOSIM (analysis of similarities) test was used to identify microorganism differences (15, 20). To identify

the indicators of each phase, we conducted the following analyses. Using R language (v.3.3.2), we conducted intergroup Venn analysis (21), drew a ternary plot of genus abundance (22), compared the indicator values (IndVal) of each sample in the group, and drew a bubble plot (23). The Welch's test was used to compare genera among groups (20). We used the PICRUSt2 (24) and BugBase (25) methods to predict the function and phenotype of the gut microbes, respectively, by comparing them with the Kyoto Encyclopedia of Genes and Genomes (KEGG) information database and drew heat maps. Finally, the SparCC (sparse inverse covariance estimation for ecological association inference) algorithm was used to analyze the changes in gut microbes in the E1 and E2 groups, and a network map was drawn (26).

## Plasma metabolite extraction and data acquisition

After preparing the correction solution of the plasma standard curve (27), blood samples for precise global metabolomic analysis were processed as follows. All samples were thawed at 4°C, and 100 µL of each sample was accurately transferred into a 2 mL centrifuge tube. Then, the mixed internal standard solution and −20°C methanol (Thermo, Waltham, USA) were added for vortex and centrifugation. The supernatants were obtained. After vacuum concentration, drying, and re-dissolution with 80% methanol solution, the supernatants were obtained by centrifugal again. Twenty microliters of each sample was taken as quality control (QC) samples (28), and the remaining samples were subjected to high-performance liquid chromatography-tandem mass spectrometry (LC-MS) analysis (Thermo Fisher Scientific, USA) (29, 30). The components were separated using chromatography, followed by mass spectrometry for data acquisition.

## Data preprocessing, relative quantification, and data inspection of plasma metabolites

The ProteoWizard software (v.3.0.8789) was used to convert the raw data obtained into mzXML format (xcms input file format) (31). R language was used for peak identification, filtration, and alignment. Targeted relative quantitative analysis obtained the original mass spectrum data of the standard curve correction solution and sample solution through the preparation of the isotope internal standard mixed, standard curve correction, and sample solution to calculate the relative quantitative results of the metabolites in the sample solution. After screening the relative quantitative analysis results, the precursor molecules were obtained under positive and negative ion conditions, respectively, and the data were exported to Excel for subsequent analysis. Batch normalization of the peak area was performed to compare the data of different magnitudes. The QC samples obtained in the preceding steps were analyzed using principal component analysis (PCA) for their dense distribution. Quality assurance (QA) removes peaks with poor repeatability in QC samples to obtain high-quality datasets (28). The proportion of characteristic peaks with a relative standard deviation of <30% should be calculated to judge the data situation in QC samples (32).

## Plasma metabolites data analysis

First, we performed a primary analysis of the data of metabolites that had yet to be identified. We applied the principles of chemometrics and multivariate statistical methods to reduce and classify data for a more intuitive and effective comparison of the multidimensional metabolome information. The standardized method adopted was the R language Ropls package based on autoscaling, mean-centering, and the scaled-to-unit variance algorithm, including PCA, partial least squares-discriminant analysis (PLS-DA) (33), and orthogonal partial least squares discriminant analysis (OPLS-DA), from which corresponding score plots were obtained (34). Differentially abundant metabolites (DAMs) were screened using parameters with variable importance for the projection (VIP) ≥1.00 and $P < 0.05$ (34). We then identified the metabolites and performed

a secondary analysis, with the exact molecular weight of the metabolites <15 ppm according to the MS/MS pattern. The accurate information on metabolites was obtained by further matching annotations in the self-built standard database of Panomic. Then, all the metabolites identified were classified by comparing them with the KEGG and Metabolon databases (35). When performing analysis of differential metabolites, the relative content of all metabolites needs to be converted into a standard score (Z-score) to measure the relative content of metabolites at the same level (36). Using the R language, we conducted hierarchical clustering analysis, drew heat maps (37), and calculated the Pearson correlation coefficient analysis between metabolite pairs to draw correlation heat maps of DAMs. Using MetPA, a part of the MetaboAnalyst (www.metaboanalyst.ca) database based on the KEGG metabolic pathway, we analyzed the related metabolic pathways of different metabolites between the groups. We drew a network map of metabolic pathways and a bubble map of the influencing factors of metabolic pathways (38).

## Correlation analysis between gut microbes and plasma metabolites

We use two models. One is the bidirectional orthogonal projections to latent structures (O2PLS) model, which was constructed using gut microbial genera and metabolite abundance data. The model used the R language OmicsPLS package to develop and output the contribution of each part, draw the loading plot, and synthesize the associated loading plot of both omics (39). The other model was the Pearson correlation coefficient model, which measures the correlation between two variables and represents the strength of the covariance of both variables. A heat map was drawn with an absolute correlation coefficient of >0.5 to present the correlation between microorganisms and metabolites, and the network map was drawn using the R language igraph package (40, 41).

## Statistical methods and data analysis software

Repeated measures analysis of variance, combined with multiple comparisons, the Welch's $t$-test, and the Mann-Whitney Wilcoxon test were used to analyze continuous variables according to the corresponding circumstances. Pearson's χ test was used to analyze categorical variables between the groups, depending on the validity of the assumptions. Pearson's correlation coefficient was performed to determine the correlation between different omics. SPSS V22.0 (Chicago, IL, USA) was used for the statistical analysis. GraphPad Prism (San Diego, CA, USA) was used to prepare the graphs. Table S1 presents the names and versions of the R language packages used in this study. Two-sided statistical significance was tested, and only $P < 0.05$ or a corrected $P < 0.05$ was significant.

## RESULTS

### Clinical features of participants

Thirty patients who underwent laparoscopic anterior resection of the rectum, ileostomy, and ileostomy closure were included in this study. The patient samples were obtained, and the entire clinical history and biochemical test data were recorded. Table S2 presents the baseline patient information. Participants had an average ileostomy duration of 179 days, so we divided the patients into a long-term (L) group (12 admissions) and a short-term (S) group (18 admissions), according to the limit of 180 days. The male-to-female ratio was 2:1, and the average age was 58. By comparing the changes in clinical indicators of patients at E0, E1, and E2, the levels of total protein, albumin, hemoglobin, red blood cell count, plasma creatinine, and blood uric acid decreased significantly at E1 and gradually returned to standard levels at E2. In addition, aspartate aminotransferase and alanine aminotransferase decreased slightly at E1 and remained at a low level at E2. In contrast, the white blood cell count increased slightly at E1 (Fig. S1).

## Altered gut microbe diversity of participants

According to the flatness of the rarefaction curve (Fig. 2A), the sequencing and samples were sufficient for taxonomic identification. Figure 2B presents four important parameters for α diversity, namely, Simpson, Shannon, Chao1, and Sob index. Overall, the richness and evenness of the patients' gut microbes were gradually increasing with time. In phase E2, 1 month after ileostomy closure, all α diversity indexes increased significantly compared with the previous phases. However, the changes in E0 and E1 were less obvious, although there are significant differences between Sob and Chao1, which focus on species richness. Figure 2C shows the β diversity between the three groups, which reflects microbial differences between individuals. The E2 group's gut microbes were relatively concentrated and rarely overlapped with those of the other two groups, meaning the intestinal environment of the patients gradually reached a state of balance 1 month postoperatively owing to routine meal feeding. In contrast, individuals in the E0 and E1 groups had scattered gut microbiomes. The ANOSIM test results revealed significant differences in β diversity between and within the three phases (Fig. 2D). We also compared the duration of ileostomy, which had little effect on the species diversity of gut microbes, suggesting that the homogeneity in the E0 group was relatively consistent (Fig. S2).

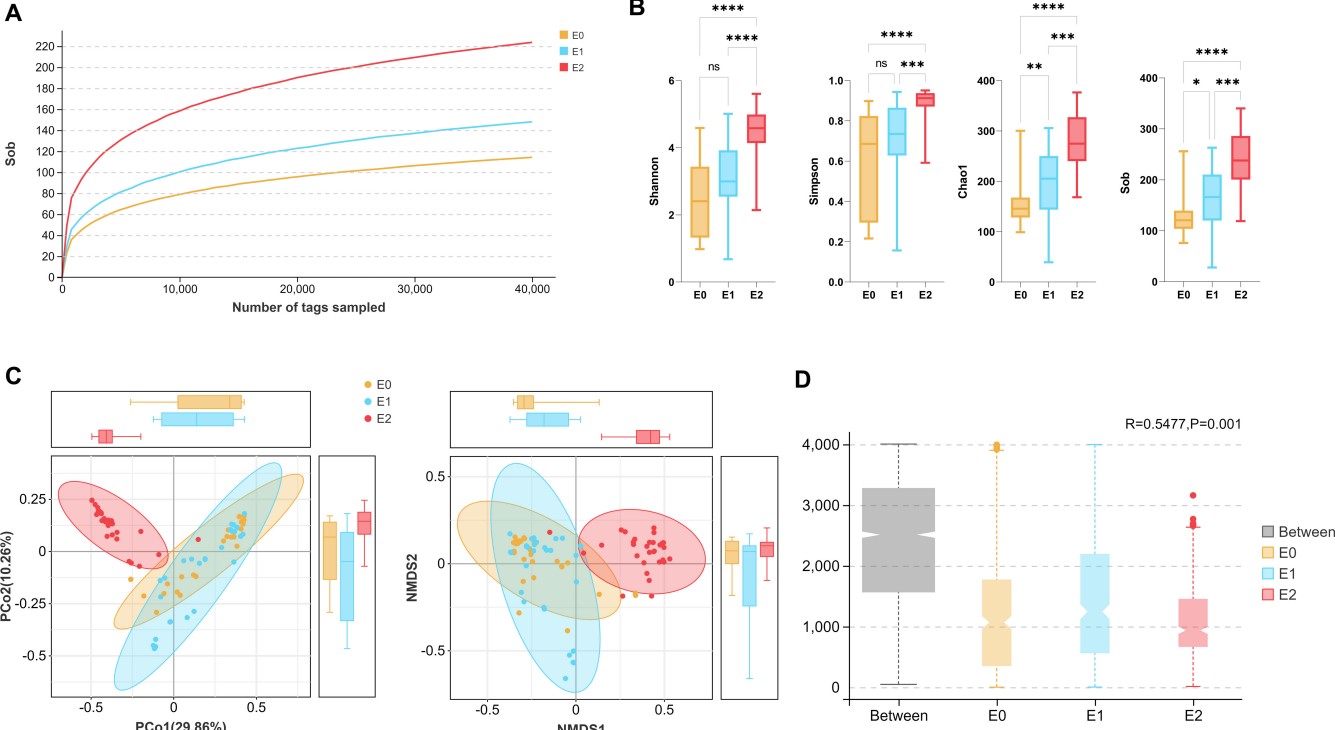

**FIG 2** Comparison of species diversity before and after ileostomy closure. (A) Dilution curve. The horizontal coordinate represents the number of tags extracted, and the ordinate indicates that the diversity index value was calculated when the corresponding number of tags was extracted. (B) α diversity index. From left to right are Shannon, Simpson, Chao1, and Sob. (C) β diversity analysis model. The left and right sides are the PCoA and NMDS analyses. (D) ANOSIM of gut microbes. $R = 0.5477$ and $P = 0.001$, indicating a moderate statistical difference among the three groups. The level of species analysis above is at the OTU, and the distance is Bray. Repeated measures ANOVA combined with the Tukey test were used in the B plot, * signifies $P < 0.05$, ** indicates $P < 0.01$, *** signifies $P < 0.001$, **** means $P < 0.0001$, ns represents no significant difference. Abbreviations: PCoA, principal coordinate analysis; NMDS, nonmetric multidimensional scaling analysis; ANOSIM, analysis of similarity; OTU, operational taxonomic unit.

## Composition, difference, and function analysis of gut microbes before and after ileostomy closure

Through the community composition analysis, the main gut microbes in each group were found. The left diagram of Fig. 3A shows that the bacteria phylum with the highest relative abundance in E0 and E1 is *Proteobacteria*, while *Firmicutes* is in E2. From E0 to E2, *Proteobacteria*, *Verrucomicrobia,* and *Patescibacteria* gradually decrease, while *Firmicutes*, *Bacteroidetes,* and *Actinobacteria* gradually increase. On the right of Fig. 3A, the relative abundance of the three groups at the genus level can be observed. The relative abundances of *Escherichia-Shigella* and *Streptococcus* at E0 were more than those at E1 and E2. When comparing the E1 and E2 groups, the relative abundance of *Enterococcus* in the colon rapidly decreased after closure surgery, while that of *Faecalibacterium*,

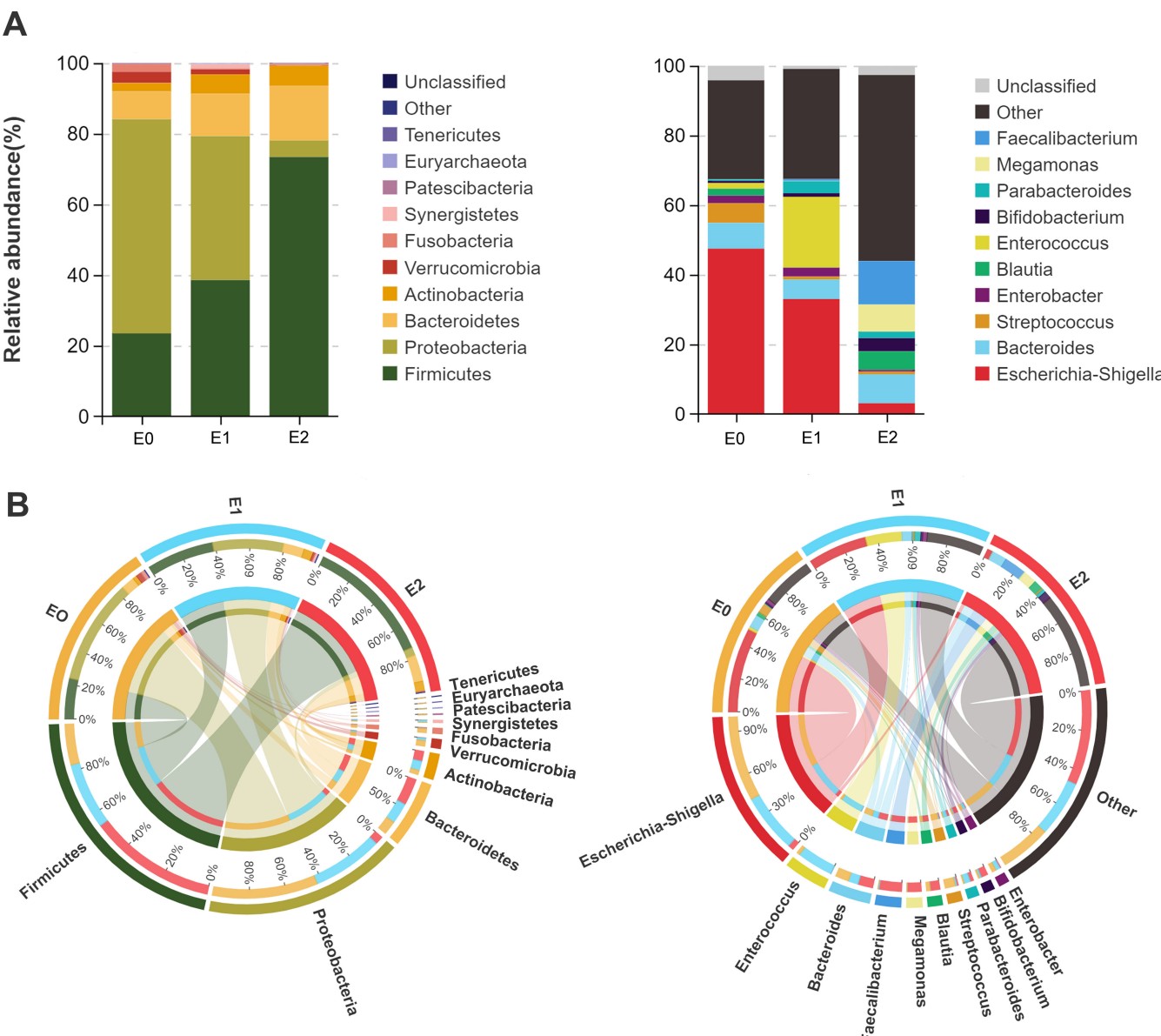

**FIG 3** Analysis of gut microbial composition before and after ileostomy closure at group level. (A) Relative abundance of gut microbes stack maps. The left diagram shows the difference in the composition at the phylum level, while the right diagram shows the genus level. The horizontal axis represents the grouping, and the vertical axis represents the relative abundance of species. (B) Circos maps of flora distribution. The left and right diagrams are the phylum and genus level Circos maps, respectively. One side of the graph is the group information, and the other side is the species information. The lines on both sides indicate a pair of corresponding relationships, and the thicker the lines, the larger the abundance value.

*Megamonas*, *Blautia,* and *Bifidobacterium* increased. From E0 to E2 groups, the relative abundance ratio of bacteria becomes increasingly complex, with other types of bacteria at genus level increasing rapidly, suggesting that the gut microbe diversity may be improving (Fig. 3B). The analysis of community composition at the sample level can be seen in Fig. S3.

According to the Venn diagram based on the abundance of bacterial genera (Fig. 4A), 81 common bacterial genera were detected in the whole three phases, while E0, E1, and E2 had 25, 25, and 30 unique bacterial genera, respectively. An enriched ternary graph

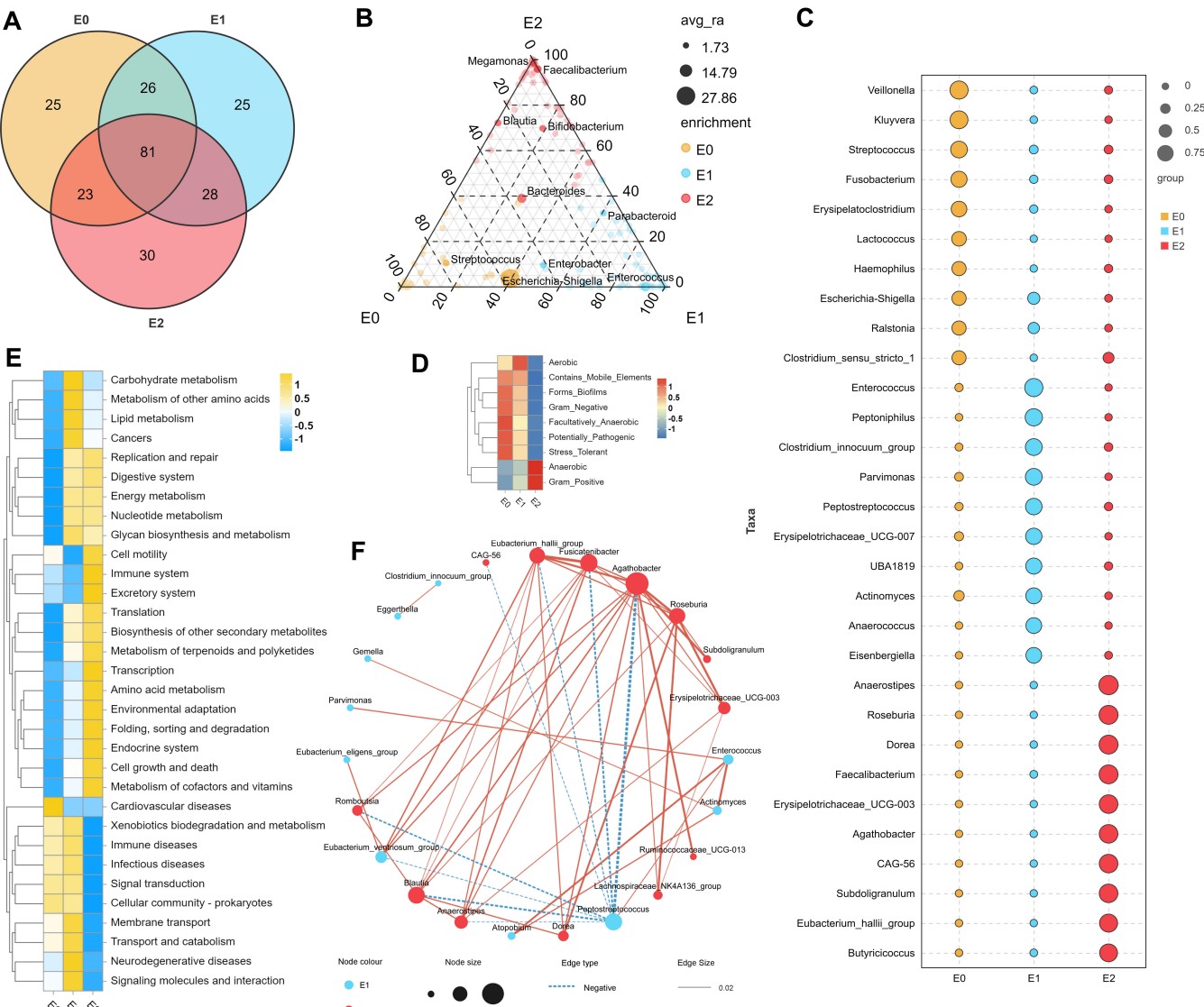

**FIG 4** Indicator flora genera and functional prediction at different phases. (A) Venn analysis plot of gut microbes. When the average tag value is > 1, the bacteria genus in the group is considered to exist. The diagram presents the number of common and unique bacteria among the groups. (B) Enriched ternary graph of gut microbes. The different dot colors represent the different enriched groups. Different dots represent different genera, and the size of the dots indicates the average abundance of the genus. The location of the points is constituted by the relative abundance share of the species in the three groups. The top 10 bacterial genera with the highest average abundance are annotated. (C) Bubble plot of IndVal analysis. This plot reveals the enrichment of the top 10 genera by IndVal in their respective periods. The color of the dots represents the different groups, and the size of the dots represents the IndVal. (D) Phenotypic abundance heat map. Nine types of gut microbes' phenotypic abundance are presented. (E) Heat map of functional abundance. Dynamically displaying the function distribution of different groups. (F) The correlation network of gut microbes between phases E1 and E2. The dot color represents the different enriched groups, and the dot size indicates the number of correlations. The dashed line represents the negative correlation, the solid line indicates the positive correlation, and the line thickness represents the strength of the correlation. Abbreviation: IndVal, indicator value.

presents the results of the genus difference test among the three groups (Fig. 4B). Most of the enriched bacterial genera showed a high proportion in their own group, indicating that the degree of difference between the groups was very significant. In the analysis of indicator bacterial genera, 88 from the three groups differed significantly, and the first 10 IndVal of each group are selected and are displayed in Fig. 4C. *Veillonella*, *Kluyvera*, and *Streptococcus* were the most significant in group E0, whereas *Enterococcus*, *Peptoniphilus*, and *Clostridium innocuum group* were the highest in group E1. *Anaerostipes*, *Roseburia*, and *Dorea* were most prominent in group E2. We also obtained the top 20 distinct genera of the gut microbes between E1 and E2 (Fig. S4A) and between E0 and E1 (Fig. S4B) via statistical analysis.

As shown in Fig. 4D, anaerobic and gram-positive bacteria gradually increase from E0 to E2, whereas gram-negative, potentially pathogenic, and stress-tolerant bacteria gradually decrease. The abundance of aerobic bacteria first increased and subsequently decreased; aerobic bacteria were most abundant in the fecal samples after ileostomy closure. The differences in functional changes shown in Fig. 4E are also significant. The activity of some signal pathways associated with diseases seemed to decrease significantly in 1 month postoperatively, while metabolism-related and functional system-related signal pathways stabilized or increased. We performed a SparCC correlation analysis of the microbes in two groups (E1 and E2) (Fig. 4C). *Agathobacter* positively correlated with the abundance distribution of up to 12 bacterial genera, while *Peptostreptococcus* negatively correlated with eight bacterial genera. In addition, *Fusicatenibacter*, *Blautia*, *Eubacterium hallii group*, and *Roseburia* all have a wide correlation, suggesting that the above bacteria may play a central regulatory role in the reconstruction of microbial community after ileostomy closure as network hub.

## Global metabolite differences and pathway association analysis before and after ileostomy closure

The plasma samples from these patients were analyzed using global untargeted metabolomics to demonstrate a complete picture of plasma metabolism during ileostomy closure surgery. PCA, PLS-DA, and OPLS-DA revealed significant differences in the metabolism of the three phases (Fig. S5). In addition, the most populated category among all classified identified metabolites was "amino acid" (35.4%), followed by "lipid" (23.6%), "carbohydrate" (15.0%), "cofactors and vitamins" (9.4%), "nucleotide" (9.4%), and "xenobiotics" (5.5%) (Fig. S6). We first use paired comparisons to screen out 110 DAMs (Fig. S7), and then, the most valuable 26 different metabolites are identified using repeated measure analysis of variance and plotted into a heat map as shown in Fig. 5A. Meanwhile, Fig. S8 and Table S3 show the details of the 26 DAMs, which cover multiple categories. Among these critical DAMs, the proportion of "carbohydrate" (23.1%) and "nucleotide" (15.4%) increased significantly compared to their blood composition, suggesting that these two types of metabolites may be more susceptible to ileostomy closure. From the above charts, we can also find that most of the DAMs decrease after closure surgery.

Combined with the VIP value in Fig. 5B, sucrose, succinic acid, and thymine are the top three metabolic substances with the most significant decreases from E0 to E2, whereas L-leucine, creatine, and 2-ketobutyric acid are the top three substances with the most significant increases. In addition to the metabolites mentioned above, the relative levels of 4-hydroxycinnamoylagmatine, ethylmethylacetic acid, alpha-ketoisovaleric acid, hydroquinone, L-fucose, L-glutamine, norepinephrine, pyroglutamic acid, pyrrolidonecarboxylic acid, and riboflavin suddenly decreased in a short time after ileostomy closure. However, metabolites, such as xanthine, (R)–5,6-dihydrothymine, cholesterol, hypoxanthine, L-rhamnono-1,4-lactone, and traumatic acid, were gradually decreased after surgery until measured 1 month later. On the contrary, in addition to the above-mentioned metabolites, the levels of se-methylselenocysteine and sphingosine also increased after ileostomy closure. Finally, metabolites named maletic acid and malonate

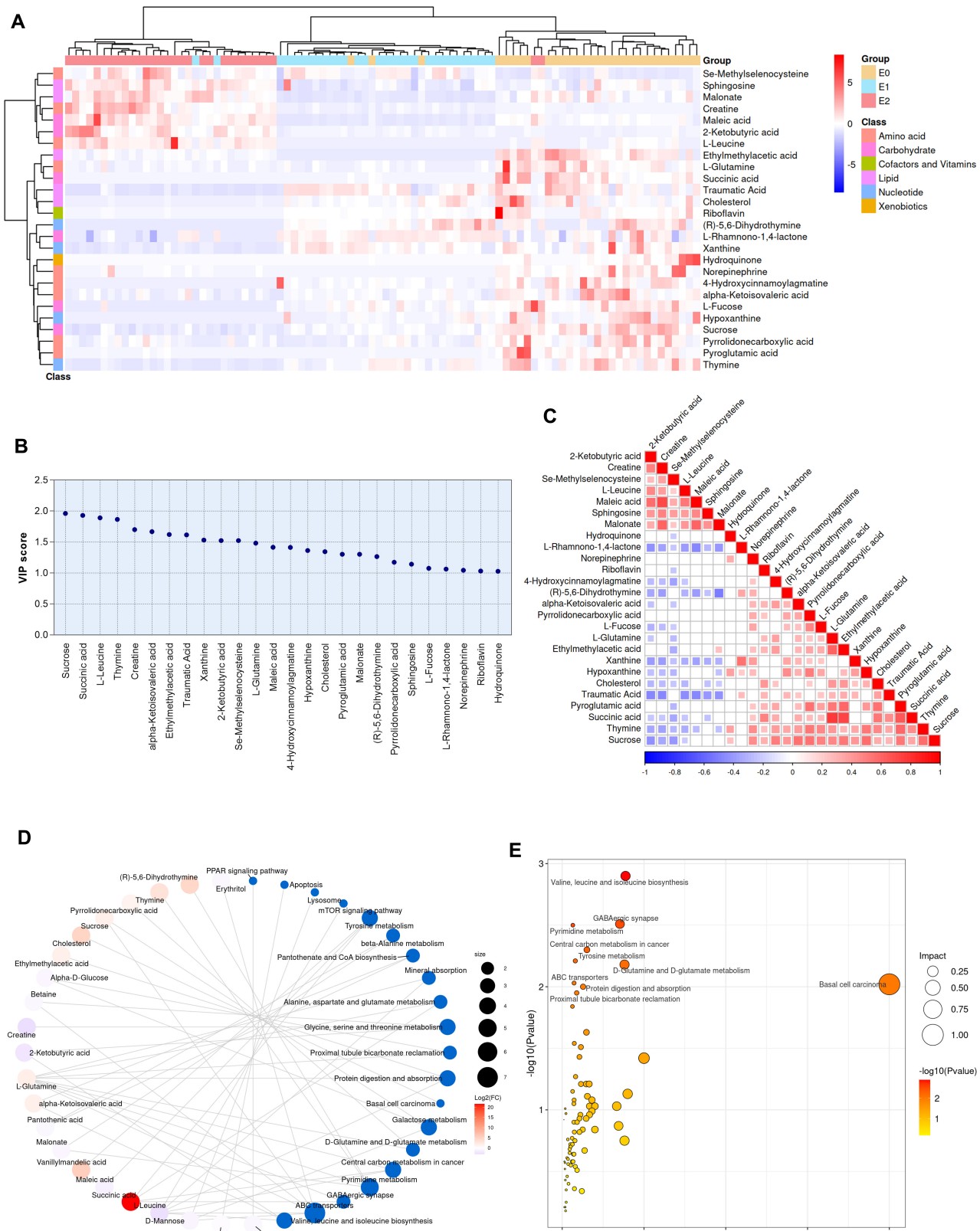

**FIG 5** Global targeted metabolites and signaling pathway association analysis. (A) Clustering heat map of DAMs. The horizontal coordinate represents a sample from each individual, and the vertical coordinate represents the name of each DAM. The color of the block represents the standardized relative abundance of the metabolite. The upper color bar represents the sample grouping. The left color bar represents the metabolite classification. (B) VIP plot of DAMs. VIP > 1.000, $P < 0.05$. (C) Correlation heat map of DAMs. Correlation analysis was conducted by calculating Pearson's correlation coefficient for two metabolites. Positive and

Fig 5 (Continued)

negative correlations are indicated by the red and blue dots, respectively. (D) Network diagram of metabolites and associated signaling pathways. Blue dots represent metabolite enrichment pathways, and other dots represent metabolites. The size of the blue dot indicates the number of molecules associated with it, and those of the other dots indicate the size of the log2(FC) value by gradients. (E) Bubble diagram of the enriched pathways of DAMs. The horizontal axis and the size of the dots represent the impact value, and the vertical axis and the color of the dots represent the $P$ value of the hypergeometric distribution test of KEGG enrichment. Repeated measures ANOVA adopted in Figure A screened DAMs. Abbreviations: DAMs, differentially abundant metabolites; VIP, variable importance for the projection; KEGG, Kyoto Encyclopedia of Genes and Genomes; FC, fold change; ANOVA, analysis of variance.

decreased immediately after surgery, but increased significantly one month later and were higher than the preoperative state.

In the correlation analysis of DAMs, we observe that "L-glutamine and succinic acid" (correlation coefficient = 0.77; $P < 0.01$), "ethylmethylacetic acid and succinic acid" (correlation coefficient = 0.73; $P < 0.01$), and "creatine and maleic acid" (correlation coefficient = 0.70; $P < 0.01$) are significantly positively correlated, while "malonate and (r)−5,6-dihydrothymine" are significantly negatively correlated (correlation coefficient = −0.52; $P < 0.01$) (Fig. 5C). Then, by comparing the information in the KEGG database, we associated the DAMs with the signaling pathway and drew a network diagram (Fig. 5D). The most abundant target metabolic pathways were ATP-binding cassette (ABC) transporters (six DAMs), followed by pyrimidine metabolism (four DAMs). The metabolite succinic acid has the highest log2(FC) value, which means that it has the strongest effect on the signaling pathway, while L-glutamine has the broadest impact as a metabolite associated with nine signaling pathways. Moreover, to investigate the disturbed metabolic pathways in patients, we performed a pathway enrichment analysis on all DAMs. The most enriched signal pathways were those of "basal cell carcinoma", "valine, leucine, and isoleucine biosynthesis", "GABAergic synapse", and "d-glutamine and d-glutamate metabolism" (Fig. 5E).

## Microbe−metabolite correlations of participants

Considering the characteristics of our experimental groups, the correlation analysis of fecal flora and blood metabolism changes in groups E1 and E2 was more homogeneous and comparable. We built an O2PLS model, evaluated its contributions, and constructed loading plots to determine the internal relationship between the two matrices (Fig. 6A). According to the figure, the gut bacteria with the strongest associations with metabolites, ranked from highest to lowest, are *Ruminococcus gauvreauii group,* *Romboutsia*, *Lachnospiraceae UCG-004*, *Dorea,* and *Turicibacter*. On the other hand, the metabolites with the strongest association with bacteria, ranked by intensity, were riboflavin, 5-aminolevulinic acid, 4-oxoglutaramate, 1-pyrroline-4-hydroxy-2-carboxylate, and citrulline. Based on the loading values of the elements, we obtained the associated load diagrams for both groups (Fig. 6B), which can more directly observe the correlation between them. In the association analysis of all the monitored gut microbes and metabolites by Pearson's correlation coefficient, there were 21 metabolites and 35 bacterial genera with strong correlation (Fig. 6C). Cholesterol was significantly negatively correlated with various bacteria, especially *Blautia* and *Faecalibacterium*, and significantly positively correlated with *Anaerococcus* and *Actinomyces*. L-leucine had the most significant positive correlation with some bacterial genera, particularly *Slackia*, *Butyricicoccus*, and *Megamonas*. We screened out the relationship between the microbes and metabolites with absolute correlation coefficients >0.5 (Fig. 6D). The number of bacteria closely associated with changes in L-rhamnono-1,4-lactone was the largest, and *Actinomyces* was most associated with the largest number of metabolites. These widely related microbes and metabolites are likely to be key points in the process of healthy reconstruction after ileostomy closure.

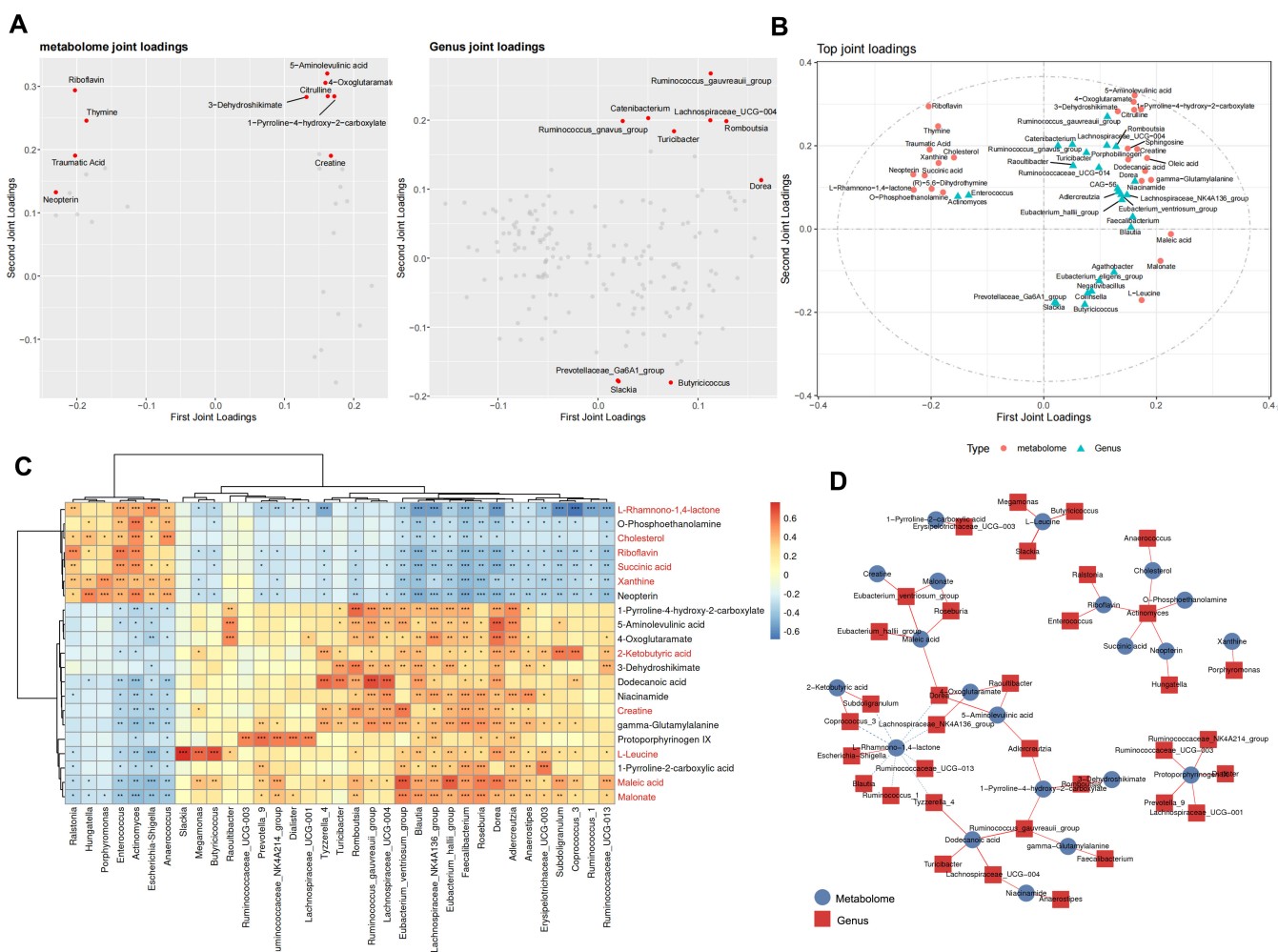

**FIG 6** Analysis of the association between gut microbial genera and plasma metabolites. (A) O2PLS loading diagram of gut microbes and plasma metabolites. Plasma metabolites are on the left, and gut microbes are on the right. The horizontal and vertical axes are the first- and second-dimensional coordinates, respectively. The greater the absolute value of the coordinates of these dots, the greater the degree of correlation. The top 10 are represented by red dots. (B) Integrated O2PLS loading diagram. The top 25 loading values (the sum of squares of loading values 1 and 2) of plasma metabolites and gut microbes were screened. (C) Correlated heat map of gut microbes and plasma metabolites. The horizontal and vertical axes are the gut microbes and DAMs, respectively. The DAMs in red font indicate that they overlap in Fig. 5A. The color gradient of the square is the size of the correlation coefficient. On the left and at the top are cluster trees for the row and column data, respectively, with asterisks in the cells indicating correlation significance $P$ values. * means $P < 0.05$, ** means $P < 0.01$, *** means $P < 0.001$. (D) Correlation network diagram of the gut microbes and plasma metabolites. The relation pairs with absolute correlation coefficient values of >0.5 are screened. The red squares and blue dots represent the gut microbes and plasma metabolites, respectively. The solid and dashed lines represent positive and negative correlations, respectively. Abbreviations: DAMs, differentially abundant metabolites; O2PLS, bidirectional orthogonal projections to latent structures model.

## DISCUSSION

In this study, through a joint analysis of microbiomics and metabolomics in 30 patients, we longitudinally described the spatiotemporal mapping of the ecological reconstruction of gut microbes and the recovery of body metabolism before and after ileostomy reversal. We analyzed the entire process through the joint dynamics of E0, E1, and E2 and observed the microbial characteristics of ileal feces before and after fermentation in the colon by comparing E0 and E1, as well as the microbial characteristics before and after microbial restoration in the colon by comparing E1 and E2. The plasma metabolite characteristics of E0, E1, and E2 reflect the metabolic changes in the body before and after ileostomy reversal and corroborate with gut microbial characteristics.

These characteristics suggest the impact of colonic abandonment on patients, further elucidating the relationship between gut microbes and metabolism in the body and providing important clues for preoperative preparation, postoperative rehabilitation, and prevention of complications in ileostomy and ileostomy reversal.

We analyzed the overall microbiome composition of individual taxa over time from three sets of samples (E0, E1, and E2). The gut microbes of patients undergo a succession of changes that correlate with a shift in the status of the colon from fecal diversion to intestinal continuity restoration and routine meal feeding. The E2 group had a significantly higher relative abundance than the E0 and E1 groups. Notably, the Simpson and Chao1 indices of the rectal samples in the E2 group were significantly higher than those in the E1 group, whereas no significant difference existed in these three indices between the E0 and E1 groups. These results suggest that the defunctioned colon of patients with ileostomy does not provide a suitable environment to maintain the diversity and abundance of colonic microbiological flora. Consequently, fecal diversion may lead to microbial loss and dysbiosis in the defunctioned colons. Interestingly, we observed a significant enhancement of intrinsic immunity in the defunctioned intestines distal to the ileostomy, which seemed to form a new homeostasis with microbial dysbiosis (42, 43). In addition, these results suggest that when intestinal continuity is restored with routine meal feeding, gut microbes recover to a standard composition within 1 month. The low α diversity in patients with ileostomy confirmed previous reports of small intestinal stoma (E0) (44) and diversion colitis (E1) (4). This trend may be explained by a combination of major alterations (fecal diversion and oxygen exposure) in the gut environment, which may provide a preferable niche for aerobic and facultative anaerobic microbes (44). Meanwhile, the individuals in the E0 and E1 groups had scattered β diversity, suggesting that species proportions within individual ileal microbial taxa fluctuate largely. These characteristics of bacterial community structure and diversity align with the previous studies on the biogeography of the human small intestine (44, 45). Indeed, the relative abundances of aerobes and facultative anaerobes were higher in ileostomy patients than in those who underwent ileostomy closure and routine meal feeding.

Due to the rapid peristalsis of the small intestine and the secretion of more bactericidal substances, the abundance of microbial species in the small intestine is relatively low. However, the role of the small intestine microorganisms cannot be ignored, which mainly includes the regulation of immune, metabolic, and endocrine functions (45). We found that the small intestine had a relatively higher abundance of *Veillonella*, *Kluyvera*, *Streptococcus*, and *Clostridium* and a lower abundance of *Bifidobacterium*. *Streptococcus* and *Veillonella* have been identified as the main microorganisms in the small intestine and are related to carbohydrate metabolism and immunity in other studies (46). Particularly, *Veillonella* has two interesting characteristics. *Veillonella*, an obligate aerobic bacterium, is closely associated with postoperative complications after ileostomy closure (47). In contrast, *Veillonella* is key in the restoration of the colonic flora after gastrectomy (48). The relative abundance of *Enterococcus* in the first postoperative stool was much higher than that of E0 and E2 groups, which was rarely mentioned in previous studies, suggesting that it was most likely present in the fecalith of a defunctioned colon. At the same time, our study unprecedentedly identified another potentially pathogenic bacterium, *Escherichia-Shigella*, which rapidly declines as intestinal function rebuilds. Regarding the key bacteria in postoperative flora reconstruction, we compared the topology of genus co-occurrence and co-exclusion networks. *Agathobacter* contributed the most positive connections (12 genera), reflecting its role as a network hub in the E2 group and its importance in reconstructing the microbial community. *Agathobacter* has been proven to be associated with the production of short-chain fatty acids, which are indispensable in maintaining the intestinal barrier (49). Functional enrichment analysis indicated that with routine meal feeding after ileostomy closure, some signaling pathways associated with disease seemed to decrease, while metabolism- and functional system-related signaling pathways stabilized or increased. These results suggest that

closure surgery should be performed promptly when the patient's physical condition is suitable.

We identified 26 DAMs after a multigroup statistical analysis of the metabolites in the E0, E1, and E2 phases, most of which have not been previously reported. First, we examined the 19 metabolites elevated during the E0 phase. Xanthine and hypoxanthine are intermediate products of purine metabolism and are closely associated with uric acid production, which could provide solid evidence for the previously reported increased incidence of kidney stones in patients who underwent colectomy or ileostomy (50, 51). In addition to chronic water and carbonate loss, uric acid imbalance may be a crucial cause of kidney stone formation in these patients (52). No significant correlation between ileostomy and gout has been observed; however, these patients may be at high risk for purine metabolism disorders. Three DAMs in the lipid metabolism pathway increased in the E0 phase, namely, cholesterol, traumatic acid, and ethylmethylacetic acid. The relationship between cholesterol and gut microbes is widely discussed, and many mechanisms have been clarified. Therefore, we considered that cholesterol increase in patients with ileostomies may be related to two factors. Firstly, cholesterol-related probiotics in contact with chyme in the colon do not work or are lost (53, 54). The correlation analysis in our study uncovered that *Actinomyces*, *Anaerococcus*, *Blautia*, and *Faecalibacterium* are strongly correlated with cholesterol metabolism and are likely to participate in regulating cholesterol metabolism. Secondly, the distal ileum ileostomy disrupts certain feedback regulatory pathways for cholesterol absorption, such as Fgf-15 expression (55). Therefore, we need to be vigilant about hypercholesterolemia in patients with ileostomies. We also found that the relative concentration of traumatic acid was higher in patients before closure surgery, and it is a long-chain fatty acid with anti-inflammatory, antioxidant, and antibacterial properties that can protect cells, reduce skin inflammation, and promote wound repair (56, 57), and the postoperative reduction may be related to the feedback of incision healing. Furthermore, we looked at amino acid metabolic pathways and identified six related substances, namely, alpha-ketoisovaleric acid, L-glutamine, pyroglutamic acid, pyrrolidonecarboxylic acid, 4-hydroxycinnamoylagmatine, and norepinephrine. Norepinephrine has been extensively studied as a neurotransmitter regulated by the gut microbes (58), which may be related to the need for neuromodulation to compensate for patients' preoperative mental and emotional stress and the reduced circulating blood volume caused by fluid loss during ileostomy (59). In addition, six DAMs were elevated after ileostomy closure. The three amino acid metabolic pathway substances are creatine, L-leucine, and se-methylselenocysteine, and their increase may be related to routine meal feeding and intestinal function recovery (60). L-leucine is an essential amino acid; however, some studies have suggested that its excessive intake may promote immune escape from CRC (61). In contrast, selenocysteine is a common organic selenium compound in food, which can help prevent cancer (62) and protect kidney function in patients undergoing platinum-containing chemotherapy (63). Other DAMs that have not yet been identified are also useful for the diagnosis, treatment, and prognosis of various diseases. In summary, the detection of these DAMs may help determine the timing of ileostomy closure and subsequent treatment.

The analysis results of the correlation between the colonic microbes and the plasma metabolites were striking, and we identified a number of strong statistically significant associations. After simple classification, we found that the relative abundance of most bacteria with positive correlation on amino acid metabolites was higher in the E2 stage. *Dorea* is associated with the largest number of metabolites and should be an important regulator in the gut, but its important role in the course of intestinal functional reconstruction has not been reported in previous studies. Although its relative abundance in the E2 period is not high (only 1.2%), its detection rate in the normal population exceeds 89% (64). *Dorea* can ferment a variety of substrates, regulate intestinal immune response by inducing Treg (65), and has a synergistic effect with *Faecalibacterium* to jointly maintain the integrity of the intestinal mucosal barrier. In addition, we unexpectedly found that *Slackia* is uniquely related to L-leucine. *Slackia* was once thought to

be a resident microbiome in the mouth and was associated with diseases, such as periodontitis (66). However, no studies have linked it to L-leucine, which may be worth further study. In our study, some DAMs of globe and microbe–metabolite correlation analysis overlap, and these metabolites can be considered to have a strong correlation with gut flora alteration and ecological restoration. In other words, their production and consumption are more likely to be influenced by the intestinal flora.

This study provides a new basis for the development of gut microbial ecology, metabolite detection, and homeostasis assessments. However, this study had some limitations. The main challenge is that the association analysis in this study was based on the results obtained from the data analysis and lacked the support from basic experiments. Secondly, we did not perform any analyses using clinical manifestations as subgroups. Finally, this study highlights the breadth of globally untargeted metabolic data analysis and the need for more precision in the individual introduction of key substances into the analysis.

## Conclusions

We described a spatiotemporal map of the intestinal microbial ecological reconstruction and metabolic recovery before and after ileostomy reversal. During temporary ileostomy, the microbes in the defunctioned colon were significantly disrupted and affected plasma metabolite levels. After the ileostomy closure, homeostasis of the internal environment quickly rebuilds and returns to a normal state within 1 month. Data analysis and a literature search revealed that these changes are strongly correlated. Therefore, a temporary ileostomy is an unnatural condition that needs to be reversed promptly when the primary disease is controlled, and key indicators are needed to guide the timing of appropriate treatment. The different gut microbes and plasma metabolites in each phase provide important clues for studying multi-omics associations and their clinical applications.

## ACKNOWLEDGMENTS

We would like to thank Professor Jingjing Xiao from the Bio-Med Informatics Research Center & Clinical Research Center of the Second Affiliated Hospital of Army Medical University for the technical support of data analysis and data visualization.

We acknowledge funding support from the National Key R&D Program of China (No. 2022YFA1304000). The project was also funded by the Discipline Outstanding Talent Project of Xinqiao Hospital (No. 2023XKRC012).

L.X.: Conceptualization, methodology, data curation, formal analysis, visualization, and manuscript preparation. X.L.: Investigation, conceptualization, methodology, data acquisition, and manuscript preparation. L.C.: Investigation, conceptualization, methodology, formal analysis, and manuscript preparation. H.M.: Methodology, formal analysis, and manuscript preparation. Y.W.: Conceptualization, data acquisition, and manuscript preparation. W.L.: Investigation, data acquisition, and manuscript preparation. A.L.: Investigation, formal analysis, and manuscript preparation. L.T.: Conceptualization, data acquisition, and data curation. X.G.: Methodology, data acquisition, and data curation. W.X.: Investigation, conceptualization, and methodology. H.Y.: Investigation, conceptualization, and formal analysis. G.J.: Funding acquisition, investigation, conceptualization, methodology, and formal analysis. Y.Q.: Funding acquisition, investigation, conceptualization, methodology, data curation, formal analysis, and manuscript preparation.

## AUTHOR AFFILIATIONS

[1]Department of General Surgery, The Second Affiliated Hospital of the Army Medical University, Chongqing, China
[2]The People's Liberation Army of China, Yunnan, China

[3]Department of General Surgery, The First Affiliated Hospital of Chongqing Medical University, Chongqing, China

[4]Department of Neurosurgery, The First Affiliated Hospital of the Army Medical University, Chongqing, China

[5]Faculty of Psychology, Southwest University, Chongqing, China

[6]Department of General Surgery, Chongqing General Hospital, Chongqing, China

## AUTHOR ORCIDs

Guangyan Ji http://orcid.org/0000-0003-4088-9636
Yuan Qiu http://orcid.org/0000-0001-5414-1827

## FUNDING

| Funder | Grant(s) | Author(s) |
|---|---|---|
| MOST \| National Key Research and Development Program of China (NKPs) | 2022YFA1304000 | Hua Yang |
| Discipline Outstanding Talent Project of Xinqiao Hospital | 2023XKRC012 | Yuan Qiu |

## AUTHOR CONTRIBUTIONS

Liang Xu, Conceptualization, Data curation, Formal analysis, Methodology, Software, Visualization, Writing – original draft, Writing – review and editing | Xiaolong Li, Conceptualization, Data curation, Investigation, Methodology, Writing – original draft | Lang Chen, Conceptualization, Formal analysis, Investigation, Methodology, Writing – original draft | Haitao Ma, Formal analysis, Methodology, Writing – review and editing | Ying Wang, Conceptualization, Data curation, Writing – original draft | Wenwen Liu, Data curation, Investigation, Writing – original draft | Anyan Liao, Formal analysis, Investigation, Writing – original draft | Liang Tan, Conceptualization, Data curation | Xiao Gao, Data curation, Methodology | Weidong Xiao, Conceptualization, Investigation, Methodology | Hua Yang, Conceptualization, Formal analysis, Investigation | Guangyan Ji, Conceptualization, Formal analysis, Funding acquisition, Investigation, Methodology | Yuan Qiu, Conceptualization, Data curation, Formal analysis, Funding acquisition, Investigation, Methodology, Writing – original draft, Writing – review and editing

## DATA AVAILABILITY

Data is included in the manuscript.

## ETHICS APPROVAL

All samples were obtained in full communication with patients, who signed informed consent forms, and were approved by the Medical Ethics Committee of the hospital (ethics number: 2020-YD056-01).

## ADDITIONAL FILES

The following material is available online.

### Supplemental Material

**Supplemental figure legends (Spectrum01191-24-s0001.docx).** Legends for Figures S1 to S6.
**Supplemental figures and tables (Spectrum01191-24-s0002.pdf).** Tables S1 to S3; Figures S1 to S8.

## Open Peer Review

**PEER REVIEW HISTORY (review-history.pdf).** An accounting of the reviewer comments and feedback.

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
