## [Reviewer comments · Microbiology Spectrum]

Microbiology Spectrum

Gut microbiome and plasma metabolome alterations in ileostomy and after closure of ileostomy

Liang Xu, Xiaolong Li, Lang Chen, Haitao Ma, Ying Wang, Wenwen Liu, Anyan Liao, Liang Tan, Xiao Gao, Weidong Xiao, Hua Yang, Guangyan Ji, and Yuan Qiu

Corresponding Author(s): Yuan Qiu, Army Medical University Xinqiao Hospital

Review Timeline:

Submission Date:	May 13, 2024
Editorial Decision:	October 24, 2024
Revision Received:	November 22, 2024
Accepted:	February 8, 2025

Editor: Wei-Hua Chen

Reviewer(s): Disclosure of reviewer identity is with reference to reviewer comments included in decision letter(s). The following individuals involved in review of your submission have agreed to reveal their identity: Qin Liu (Reviewer #1); Song Liu (Reviewer #2); Yang Yang (Reviewer #3)

Transaction Report:

DOI: <https://doi.org/10.1128/spectrum.01191-24>

Re: Spectrum01191-24 (**Gut microbiome and plasma metabolome alterations in ileostomy and after closure of ileostomy**)

Dear Prof. Yuan Qiu:

Thank you for the privilege of reviewing your work. One expert has commended on your manuscript. Below you will find his/her comments and instructions from the Spectrum editorial office for submitting a revision.

Revision Guidelines

Sincerely,
Wei-Hua Chen
Editor
Microbiology Spectrum

Reviewer #1 (Comments for the Author):

The manuscript presents valuable insights into the changes in gut microbiome and plasma metabolome in patients undergoing ileostomy and its closure. The study is well-designed and executed, with clear and concise data presentation. The research contributes to the current understanding of the temporal and spatial dynamics of the gut microbiome in such patients. The manuscript is well-organized, and the results are presented clearly. Here are some specific comments:

1. The section on the results of the metabolomics lacks some summarization. The current metabolic analysis data appears to be rather fragmented. There is a noticeable lack of comprehensive summaries comparing the changes in specific metabolites such

as lipids and amino acids. Providing detailed comparative analyses of these individual components would greatly enhance the clarity and depth of the study's findings.

2. The authors should discuss the novelty of their findings in comparison to existing literature more explicitly. For example, what are the implications of the microbes-metabolites correlations? Is it consistent with previous findings?

3. Are there any connections between Figures 5 and 6, and are there any metabolites that overlap?

Identifying those differential metabolites and being able to explain them from the differences in the microbial community could be meaningful.

Manuscript Title: Gut microbiome and plasma metabolome alterations in ileostomy and after closure of ileostomy

Manuscript Number: Spectrum01191-24

Dear Editors and Reviewers:

We want to take the opportunity to thank the editors and reviewer for reviewing this manuscript. We have carefully edited the manuscript according to the suggestions. The changes that we have made include:

1. We added more summarization of metabolites in the paragraph titled *Global metabolite differences and pathway association analysis before and after ileostomy closure* in the results section, which is marked in red font.

2. In paragraphs 2nd, 3rd, 4th and 5th of the discussion section, we refined and updated the presentation method and content, emphasizing the similarities and differences between this study and previous studies, and some new relevance statements have been added in the 5th paragraph. The above revisions are also marked in red font in the corresponding location.

3. To better represent and match the revisions, we have made some changes to Figure 6C and Table S3.

4. Corresponding authors have reconfirmed the current status of all authors and revised some information.

A detailed response to each comment of the reviewer is given below, in the remainder of this response letter.

Best regards

Reviewer #1: The manuscript presents valuable insights into the changes in gut microbiome and plasma metabolome in patients undergoing ileostomy and its closure. The study is well-designed and executed, with clear and concise data presentation. The research contributes to the current understanding of the temporal and spatial dynamics of the gut microbiome in such patients. The manuscript is well-organized, and the results are presented clearly. Here are some specific comments:

1. The section on the results of the metabolomics lacks some summarization. The current metabolic analysis data appears to be rather fragmented. There is a noticeable lack of comprehensive summaries comparing the changes in specific metabolites such as lipids and amino acids. Providing detailed comparative analyses of these individual components would greatly enhance the clarity and depth of the study's findings.
2. The authors should discuss the novelty of their findings in comparison to existing literature more explicitly. For example, what are the implications of the microbes-metabolites correlations? Is it consistent with previous findings?
3. Are there any connections between Figures 5 and 6, and are there any metabolites that overlap? Identifying those differential metabolites and being able to explain them from the differences in the microbial community could be meaningful.

Our response #1: Dear reviewer, thank you very much for your valuable suggestions, which greatly help to expand the depth and rigor of our manuscript. We have discussed your specific comments very carefully, revised the manuscript in order of number and explained the contents as follows:

1. In the fourth part of the results, the section titled *Global metabolite differences and pathway association analysis before and after ileostomy closure* has added the metabolomic summary content, which provides a comprehensive view of the changes in specific categories. According to this, the content of Table S3 is modified, the columns of *Changes* are added which represents the changed state of the metabolites after closure surgery, and the *Class* and *VIP* values of metabolites are sequentially refilmed, which can more clearly show the logic of data.

2. In the discussion section, we embellished the manuscript to highlight the novelty of our findings compared with previous researches. First of all, in the first paragraph, we once again made a minimal summary of the grouping and comparison of the whole paper. In the second paragraph of the discussion, we jointly explain the overall change and diversity of bacteria based on experimental results, clinical practice and previous studies. While in the following two paragraphs of the discussion, we analyzed some specific bacteria genus and plasma metabolites with changes respectively, and compared them with previous studies of others. We then tested microbes-metabolites correlations using computer software to find potential connections for future research, which are based on algorithms and do not necessarily match reality exactly. These revisions are marked in red in the discussion section. Because the global analysis adopted in this paper is too extensive, limited by the length of the article, we only select a part of the outstanding content in the discussion section to explain.

3. A, B and C in Figure 5 were obtained statistically based on relative abundance, and the main method of comparison was through repeated measurements of E0, E1 and E2 stages. D and E in Figure 5 are the path associations obtained by tracing the database, including KEGG. On the whole, the results of this part are based on direct reasoning. Figure 6 is mainly based on the

algorithm derivation of intestinal bacteria and plasma metabolites in E1 and E2 stages, and the result is deduced by computer software. The results are indirect and need to be verified by experiments in the future. There are overlapping metabolites between the two figures, and we can surmise that these overlapping metabolites are more profoundly affected by changes in intestinal flora. The overlapping DAMs can not only serve as a sensitive indicator of intestinal homeostasis reconstruction, but also guide the gut bacteria genus screening for clinical treatment operations such as monitoring, diagnosis and transplantation, and based on this, the potential resident probiotics in the intestine can be studied. The above ideas are logically derived and do not belong to experimental conclusions. Therefore, we can put these ideas forward in this response letter, but adopt a more cautious expression in our manuscript. According to your comments, we have revised the content of the fifth paragraph of the discussion section and marked these changes in red. Meanwhile, we marked the overlapping metabolites in Figure 6A in red and described its meaning in the figure legend.

Re: Spectrum01191-24R1 (**Gut microbiome and plasma metabolome alterations in ileostomy and after closure of ileostomy**)

Dear Prof. Yuan Qiu:

I apologise for the long delay due to the holidays. Your manuscript has been evaluated by two external reviewers. It is their consensus that your revision is acceptable for publication. Congratulations!!

I am forwarding it to the ASM production staff for publication. Your paper will first be checked to make sure all elements meet the technical requirements. ASM staff will contact you if anything needs to be revised before copyediting and production can begin. Otherwise, you will be notified when your proofs are ready to be viewed.

Sincerely,
Wei-Hua Chen
Editor
Microbiology Spectrum

Reviewer #2 (Comments for the Author):

I would like to express my congratulations to the authors for their work. The authors focus on a specific clinical issue, and successfully demonstrate a spatiotemporal map of microbial and metabolic profiles during ileostomy reversal. Their results provide first-line evidences for understanding the change caused by ileostomy that was widely applied in clinical practice. Overall, this paper has clearly answered important but ignored questions from surgeon's perspective. The authors has responded all issues raised from the original reviewers.

Reviewer #3 (Comments for the Author):

The pre reviewer effectively summarized the key points as questions, helping the author optimize the structure and readability of the manuscript. The viewpoints obtained from the dynamic study of changes in gut microbiota and metabolites before and after ileostomy in this article are very novel. The massive amount of data attached in the manuscript will help promote the

development of the field of gut microbiota research. The method is rigorous, the conclusion is scientific. I agree to accept it.

The pre reviewer effectively summarized the key points as questions, helping the author optimize the structure and readability of the manuscript. The viewpoints obtained from the dynamic study of changes in gut microbiota and metabolites before and after ileostomy in this article are very novel. The massive amount of data attached in the manuscript will help promote the development of the field of gut microbiota research. The method is rigorous, the conclusion is scientific. I agree to accept it.